# Omnidirectional Haptic Stimulation System via Pneumatic Actuators for Presence Presentation

**DOI:** 10.3390/s23020584

**Published:** 2023-01-04

**Authors:** Shogo Yoshida, Haoran Xie, Kazunori Miyata

**Affiliations:** School of Knowledge Science, Japan Advanced Institute of Science and Technology, Nomi 923-1292, Japan

**Keywords:** pneumatic, pressure, haptic, direction, presence, interaction

## Abstract

Recently, remote meetings and work-from-home have become more common, reducing the opportunities for face-to-face communication. To facilitate communication among remote workers, researchers have focused on virtual space technology and spatial augmented reality technology. Although these technologies can enhance immersiveness in collaborative work, they face the challenge of fostering a sense of physical contact. In this work, we aimed to foster a sense of presence through haptic stimulation using pneumatic actuators. Specifically, we developed a choker-type wearable device that presents various pressure patterns around the neck; the pattern presented depends on the message the device must convey. Various combinations of haptic presentation are achieved by pumping air to the multiple pneumatic actuators attached to the choker. In addition, we conducted experiments involving actuators of different shapes to optimize the haptic presentation. When linked with a smartphone, the proposed device can present pressure patterns to indicate incoming calls and notifications, to give warning about an obstacle that one who is texting might miss while walking, and to provide direction to a pedestrian. Furthermore, the device can be used in a wide range of applications, from those necessary in daily living to those that enhance one’s experience in the realm of entertainment. For example, haptic feedback that synchronizes with the presence of a singer or with the rhythm of a song one listens to or with a performer’s movements during a stage performance will immerse users in an enjoyable experience.

## 1. Introduction

Due to the decreasing opportunities for one to engage in face-to-face communication, problems such as loneliness and feeling of isolation have become more severe. These problems are somehow addressed by remote communication, which is commonly used in remote work and online gatherings. However, remote communication does not provide a sense of physical proximity and cannot recreate the communication that involves various elements, such as body heat, smell, or physical contact. In other words, the existing remote communication process cannot sufficiently relieve loneliness. Loneliness and isolation can lead to various problems, such as depression and lack of exercise, and they can reduce the quality of people’s lives [1,2]. Attempts have been made to address the above issues, including the use of robots [3,4,5,6] and virtual reality (VR) [7,8,9,10], to promote healing and to foster interaction among individuals. For example, Avatar Work [3] used OriHime-D to facilitate a physical work in the user’s place. Fribo [4] analyzes the sounds of residents’ lives as anonymous information and recognizes users’ activities. By sharing this information with friends, users can improve their sense of social connectedness. Social VR [7] describes the importance of immersive remote communication tools, such as AlterSpaceVR, VRChat, and Mozilla Hubs. Such extension of users’ experience is expected to provide a sense of presence, promote healing, and help improve one’s quality of life. However, they cannot recreate the physical contact that is necessary to promote complete healing. Haptic feedback can effectively recreate physical contact [11,12,13,14]. Mostofa et al. developed a headband haptic presentation device that provides a surrogate method of physical contact for isolated patients [11]. When the two-dimensional touch pattern is applied on the forehead of an isolated patient, the patient can feel the sensation of having their own forehead being touched. Sharing Heartbeat [12] recreates physical contact by reproducing and sharing heartbeats as haptic sensations. The heartbeat of a physically contacted user tends to synchronize with the heartbeat of the other user. The use of the device proposed herein can also potentially simulate physical contact, which could strengthen the bond between people, as well as reduce stress and other emotional and psychological discomfort.

Most of the devices described above are complex and have a rugged appearance, causing users to feel anxious and experience discomfort when wearing them. As a result, they may be not quite suitable for fostering a sense of presence or for relieving loneliness. In addition, haptic presentation involving the use of rigid materials or electricity may harm users. By contrast, pneumatic actuators are compact and lightweight and they feel soft when comes into contact with the user’s skin; importantly, they do not cause harm. On this basis, we fabricated a haptic presentation device equipped with pneumatic actuators.

In some works, haptic sensations were applied on various body parts, such as arms, body, and hands, with the use of wearable devices [15,16,17,18,19]. Haptic presentation on the neck as a choker-type device can always present the same directional information regardless of the user’s direction of orientation. For example, if the user wears a haptic presentation device on the arm and bends or twists the arm, as in David et al.’s study [19], the direction information that should be presented and the direction information presented by the device will differ. Another problem is that it may interfere with the task at hand. However, haptic presentation on the neck makes it possible to always provide appropriate directional information without interfering with daily activities. A haptic presentation device may be concealed when worn as fashion items, such as chokers, turtlenecks, and collared shirts. There is also no need for the user to remember how to operate the proposed device. In addition, the proposed device can be used daily because it can be used as a fashion item.

In this study, we investigated the effectiveness of a proposed choker-type haptic presentation device equipped with a pneumatic actuator in relieving loneliness [20] (Figure 1). The proposed system aims to present a sense of presence by haptic presentation direction, and we conduct user guidance experiments to confirm whether the user is appropriately aware of any given direction. Moreover, we investigated a sounder haptic presentation method, as well as determined the usefulness of the proposed device by conducting haptic presentation experiments involving pneumatic actuators of different shapes. The four actuators made of polyethylene could apply equal amounts of pressure around a user’s neck. In addition, we have fabricated a small, lightweight, and safe haptic presentation device that uses air pressure to present sensations. When wearing the choker, the user could feel haptic sensations around the neck, and this design does not cause any discomfort. The user can sense the presence in each direction by presenting pressure using the proposed system. The pneumatic actuators are agile and capable of sequential pressure presentation. Therefore, it is also possible to express haptic sensations such as heartbeat to sense a presence of a person. In addition, the manner and the order by which actuators are inflated may be varied to indicate various situations, such as to provide notice about an approaching obstacle or about an incoming notification in smart devices, to indicate when to start and to stop walking, and to indicate whether haptic presentation devices are linked to VR applications. Furthermore, once navigation guided only by pressure presentation patterns instead of screen displays or voice guidance is made possible, users will avoid searching for directions using their smartphones while walking and will refrain from using such a navigation system in noisy environments.

## 2. Related Works

This section presents the related studies on devices that present haptic sensations through pneumatic pressure.

### 2.1. Wearable Devices

In the field of human augmentation, extending human physical capabilities with the use of wearable devices has been widely proposed. For example, Xie et al. developed a physical tail that can support the user’s weight [21]. The proposed device was also designed to indicate one’s emotions, which is manifested in the tail movement. xClothes uses a stretchable structure to extend the human body temperature sensation [22]. It uses an open/close hole structure in a piece of clothing; this structure can be controlled using servomotors and wires. This device improves ventilation by controlling the hole size when the user’s body temperature rises, thereby supporting thermoregulation. However, reinforcing user’s senses is challenging. Jeong et al. developed a thermal stimulation device using Peltier elements worn on the wrist and on the back of the neck as a messaging tool [23]. Yamazaki et al. proposed a neck-worn haptic device that can dynamically modulate the amplitude, phase, and frequency of vibration, and they conducted experiments using only haptic information to reach an invisible image [24]. However, in the former case, the device must be secured on the back of the wrist or neck to present thermal stimuli, and the pressure it applies may cause discomfort. In the latter case, the system only grasps distance and direction from the vibrations presented on either side of the neck; an application linking the system to VR applications or the ability to guide the user to a destination is lacking. ActiveBelt [25] presents intuitive directional information to the user through vibration stimulation generated by a vibrator attached to the belt. However, because the device is embedded in a piece of clothing, the vibration stimulus may not be easily transmitted to the user depending on the thickness of the clothing. Shaking The World [26] uses galvanic vestibular stimulation to present a virtual acceleration sensation to the user. It also enables pedestrians to deviate from their normal straight-line path. This device provides guidance to a pedestrian through radio control, allows automatic collision avoidance, and supports GPS-based navigation while walking. However, the safety of the system for long-term wearing has not been investigated.

### 2.2. Pneumatic Actuators

Niiyama et al. proposed a sticky actuator that reproduces motions, such as rotation and deflection [27]. Free-form pneumatic actuators, such as rectangular, circular, and ribbon-shaped actuators, may be produced from inexpensive plastic sheets. Fabricated pneumatic actuators have been applied in the flapping of paper cranes and in the movement of robot arms and legs. However, it has not yet been investigated as a haptic presentation device in humans. Sonar et al. devised a wearable skin-like interface that can reproduce the roughness, shape, and size of an object by using a small pneumatic actuator [28]. This system uses a soft pneumatic actuator (SPA)-based skin and a marker attached to the user’s finger; the SPA skin is then read by a camera to determine the position of the user’s finger. When the user’s finger approaches a preset contour, the system applies pressure to the finger. In this way, the user can recognize the shape of a circle or a trapezoid. PneuMod [29] is a pneumatic/thermal haptic presentation device. With the use of Peltier elements and air pressure-activated silicon bubbles, thermal air pressure fade-back can be presented through shape, position, pattern, and motion effects. The temperature, the rate of expansion of the bubbles, and their degree of expansion can also be controlled. This device can be attached to the user’s socks or arm covers. Delazio et al. proposed the Force jacket, which applies varying amounts of pressure using pneumatic actuators embedded in a jacket [30]. By inflating an airbag inside the jacket using an air compressor and an air vacuum, the user can feel various sensations, such as the impact of being hit by a ball or the feeling of being wrapped up by a snake. However, Force Jacket uses a large air compressor to feed air to the actuators, consequently limiting the mobility of the user. 

The above devices cannot be used as wearable devices because of their numerous components and their complex structure.

### 2.3. Haptic Stimulation

HeadBlaster [31] is a research project that uses air pressure to continuously present a sense of acceleration to the user. A sensation of sustained acceleration or motion can be created with the application of sustained force. Given that only a small amount of force must be applied to the head, a wearable design that is not tied to a large mechanical motion platform may be created. This improves mobility and supports room-scale VR experiences. However, prolonged use is not advisable due to VR sickness and due to the weight of the device. Tactile presentation technology not only improves immersion in virtual and augmented reality, but it also contributes significantly to the reproduction of haptic perception in real space. MudPad [32] is a system that enables localized active haptic feedback on a multi-touch surface. With the use of ferrofluid and electro-magnets, the system can support instantaneous motion. Cross-Field Haptics [33] proposes a new method of drawing haptic sensations using ferrofluids and electrostatic fields. It reproduces the haptic sensation of a projected texture using the flexibility of ferrofluid and the resistance force of electrostatic adsorption caused by an electric field. In addition, it can display various textures by simultaneously combining different forces, such as pulling and pushing forces. However, its application is limited by the materials used in the device and by the method used to present haptic sensations.

Table 1 shows a comparison of the proposed device with some of the related work introduced. On the basis of the related works, we aim to develop a device that applies equal amounts of pressure to the entire circumference of the neck through the inflation of pneumatic actuators. The proposed device can be used for various applications. This study focuses on the presentation of directions to a walking user. By inflating an arbitrary actuator and fostering a sense of presence in each direction the user must take, we assess the usefulness of the basic function of the device, which is to arouse attention. In addition, we conducted evaluation experiments on the wearing comfort of the device, the degree of pressure, and the differences in the user’s perception of pressure as a function of actuator shape. We confirm the usability of the device and discuss herein its future applications.

## 3. Proposed Device

The proposed device presents haptic sensations around the user’s neck using actuators that are directly attached to the inner side of a choker. Once the actuator attached to the left side of the choker is activated for example, the user can feel pressure on the left side of the neck. In this way, the choker directs the user’s attention toward that direction or to any other direction. In addition, the actuators may be activated individually or sequentially and thus the device can present various haptic patterns.

The advantage of using pneumatic pressure is that the user can feel the pressure from the actuator even if the actuator is not in close contact with the skin. By contrast, devices that use thermal stimuli to provide tactile stimulation to a user [34] or that produce electrical stimuli to mimic the weight of an object and consequently felt by the user in virtual space [35] do not use air pressure. Such devices or sensors must be in close contact with the skin, which may cause squeezing and discomfort. As for pneumatic actuators, they increase in volume when air is fed into them. Thus, the user can feel a haptic sensation from the actuator even if the actuator is not in close contact with the skin.

### 3.1. Fabrication of Pneumatic Actuator

Figure 2 shows the proposed pneumatic actuator. When the actuator is supplied with air, it inflates to a thickness of about 0.8 cm; as a result, the user can feel a haptic sensation (Figure 2a). The actuator is made of 0.08 mm-thick polyethylene film, cut into a rectangular shape measuring 2.0 cm × 1.5 cm, and is supplied with air by a silicon tube (Figure 2b). In this study, silicone tubes with outer and inner diameters of 3.0 and 2.0 mm, respectively, and silicone tubes with outer and inner diameters of 6.0 and 4.0 mm, respectively, were used, depending on the shape of the syringe, on the silicone tube, and on the connector connecting both elements. 

We used a soldering iron to seal the film [27]. Figure 3 shows the fabrication of a pneumatic actuator. First, we drew a rectangular outline (2.0 cm × 1.5 cm) on a polyethylene film. Next, we placed a cookie sheet over the polyethylene film. Finally, we traced the rectangular shape on the cookie sheet with a soldering iron, producing a pneumatic actuator. After bending the end of the tube to prevent air leakage, we created a 0.1 cm × 0.1 cm hole along the side of the tube. We also created a 0.1 cm × 0.1 cm hole at the center of one side of the actuator; the actuator and the tube were glued to each other using *Aron Alpha for plastic*, ensuring that both holes are perfectly aligned.

### 3.2. Prototype

Figure 4a shows the prototype of the navigation device consisting of pneumatic actuators. Four actuators mounted at equal intervals around the neck can apply pressure in eight directions. For example, activation of the two front actuators indicates forward direction, and activation of one of the actuators located on the forward diagonal left side indicates a diagonal movement towards the left side. To improve wearing comfort and stability, we covered the actuators with a soft fabric (Figure 4b). The proposed method assumes that each actuator is controlled by an electronic device that automatically delivers air. However, this study aims to create a prototype and only check the feel of the actuator; thus, the actuators were manually operated using a syringe. The choker used is a standard commercial product with a length of 40.0 cm and a thickness of 1.5 cm (Figure 4c).

## 4. Preliminary Study

Before conducting the user study, we first conducted two preliminary studies. One investigated the relationship between the direction of pressure presentation and the direction of the perceived stimulus. The second investigated the differences in the stimulus perceived by the users as a function of the shape of the pneumatic actuators.

### 4.1. Preliminary Study 1: Haptic Presentation of Multiple Pressure Patterns

The direction of the stimulus perceived by the users may differ depending on the number of actuators and their location. Therefore, we verified the relationship between the direction of pressure presentation and the perceived stimulus direction. Specifically, we applied 15 pressure patterns (Figure 5a) to the users and asked them to identify the direction each applied pressure comes from. After the preliminary study, we set up eight pressure presentation patterns, as shown in Figure 5b. These patterns direct the user to eight directions. In addition, activation of all actuators directs the users to stop walking. The appropriate combination of actuator motions was based on the results presented in Figure 5c. This experiment involved seven subjects aged 20–30 years, six of whom were males.

### 4.2. Preliminary Study 2: Haptic Presentation by Actuators with Different Shapes

To investigate the differences in the users’ perception of pressure as a function of actuator shapes, we conducted experiments involving four shapes of pneumatic actuators. In addition, we modified the tube attachment to reduce the thickness of the actuators when inflated (Figure 6). Each actuator shape had a standardized area of approximately 4.0 cm^2^. This experiment involved eight subjects aged 20–30 years, of whom four were males. In the experiment, each user was presented with four different shapes of actuator. For each shape, we presented eight pressure patterns (Figure 5b), and we asked the users to indicate which actuator was activated and from which shape did they perceive the greatest pressure. Each correct answer was equivalent to one point. This means that the number of correct answers was scored on an 8-point scale for each shape.

Table 2 shows the number of correct answers as to which actuators were inflated; the same results are graphically presented in Figure 7. In Table 2, *F*_ave_ and *M*_ave_ are the average scores for the female and male respondents, respectively, and *Ave* is the average score for all respondents. In addition, the data were subjected to Wilcoxon’s rank-sum test to ascertain whether a significant difference exists between the two shapes with the largest difference in terms of the number of correct answers or whether a significant difference exists in the number of correct answers as to which actuators (based on shape) were activated, regardless of the gender or age of the respondents (Table 3 and Table 4).

Table 3 shows the average number of correct answers as to which actuators were inflated; *C*_ave_ and *R*_ave_ are the average numbers of correct answers for the circular and rectangular actuators, respectively. Table 4 shows the distribution of correct answers by gender. In Table 3 and Table 4, *z* and *P* are the test statistic and *p*-value, respectively. The significance level was set at α = 0.05. Table 3, the difference in average values was small, and no significant difference was observed between the two shapes. This suggests that the pressure sensitivity toward differently shaped actuators did not significantly differ. This result may be attributed to the small size of the actuators, wherein there may not have been much difference in their degree of inflation. In Table 4, no significant gender differences were observed both in terms of average and *p*-values, suggesting that there is no gender difference in pressure sensation. The actuator that was easiest to identify when inflated was the circular actuator, which was selected by six users. No users selected the rectangular actuator, and this result is consistent with the number of correct answers presented above. While there is no significant difference in the number of correct answers in terms of actuator shapes, most users indicated that the circular actuator presented pressure most clearly. However, using the shape that can present the greatest amount of pressure to the user does not necessarily improve the accuracy of direction presentation. Future works aiming to improve the actuator should thus focus not on changing the shape but on adjusting the degree and speed of inflation.

## 5. User Study

The prototype of the proposed device can be worn around the user’s neck. We verified the wearing comfort of the proposed system and the effectiveness of the pressure and direction presentation. We conducted movement experiments, wherein one has to reach a certain destination with only the pressure presented by the device as guide (Figure 8). Whenever the user approaches a turn, we present pressure to indicate the desired direction (Figure 9). The pressure was presented to the user multiple times at a time. After facilitating the walking experiment, we conducted a second experiment to verify the difference in the user’s perception of pressure caused by differences in the actuator shape.

### 5.1. Guided Walking Experiment

We conducted an experiment involving nine subjects aged 20–30 years, seven of whom were males. The user study consisted of an experiment where the user must take a route while being guided by the proposed device, followed by a questionnaire survey. The experiment was conducted using the Wizard of Oz method [36] wherein the actuator is manually operated. When a user must take a turn, the experimenter activates a pneumatic actuator using a syringe to direct the user. Being not informed of the destination and wearing a noise-canceling earphones that block out external sounds, the user relies solely on the pressure presented by the proposed device to determine the route leading to the destination. The start and end signals are also given as a pressure pattern. The start signal immediately indicates the direction to take, whereas the end signal (i.e., pattern No. 15 in Figure 5b) was activated once the user reached the target destination. After the experiment, we conducted a questionnaire survey to evaluate the usability of the system, the feeling of wearing the device, and the presentation of pressure. The following questions were asked, each of which could be answered using a five-point Likert scale (5: strongly agree; 1: strongly disagree).

Did you clearly feel the pressure presented by the actuator?Did you clearly feel the differences between directional presentations?Do you feel any discomfort due to the pressure presentations?

### 5.2. Result of the Guided Experiment

Eight of the nine users reached the target destination while being guided only by the proposed device. The users had no idea about the starting point and the route going to the destination. In fact, when the users arrived at the starting point, some of them were facing different directions relative to the destination. However, the start signal guided all users to start moving in the direction leading to their destination. In addition, the users were able to recognize the stop signal and were able to complete the experiment with only the device as their guide. This result suggests that pneumatic actuators, which could present pressure around the neck, could be useful components of a navigation system. Navigation involving the use of a screen display or voice guidance in mobile devices are associated with such problems as the need to use a smartphone while walking and the need to survive the challenges posed by noisy environments. These problems, as suggested by the user study, could be addressed by the proposed device. Meanwhile, one subject was unable to reach the destination, as the subject mistook the backward pressure as the stop signal. Therefore, it is necessary to investigate and analyze the amount of pressure and the presentation position suitable for each user. It is also necessary to improve the actuator to enable a greater number of subjects to feel the pressure position evenly.

### 5.3. Result of the Questionnaire Survey

Figure 10 shows the questionnaire survey results. The survey items evaluated the differences in direction presentation, the degree of pressure presented by the pneumatic actuator, and the level of comfort when wearing the device. The results confirmed that all subjects “felt the pressure” during the pressure presentation. Perception of different presentation directions were also reported by most of the subjects; only one subject “did not feel much difference”. The reason for this may be that the subject’s neck was too thin that the actuators did not rest on the neck. It is also possible that the actuators were hitting a point where haptic sensations were difficult to perceive. This problem could be solved by inflating the actuators further and by increasing the contact area. Regarding discomfort when wearing the device, only four subjects indicated that they experienced “no discomfort”; thus, the majority of the users experienced discomfort when wearing the proposed device. It is possible that some users feel uncomfortable wearing something around their neck, as the edge of the actuator irritates the user’s neck and because the silicone tubing connected to the actuator renders the wearing of the device cumbersome. A softer cloth must be placed between the neck and the device to avoid causing pain, the edges of the actuator must be smoothened, and thinner silicone tubing must be used to render the device unnoticeable.

## 6. Summary

We propose herein an omnidirectional pressure presentation system involving pneumatic actuators. We fabricated a prototype wherein we attached rectangular pneumatic actuators (2.0 cm × 1.5 cm) to the inner side of a commercially available choker; the actuators were inflated in order to present pressure to the user’s neck. By conducting an experiment with a focus on giving direction to the walking users, we investigated the difference in pressure perception and we determined the usefulness of the pneumatic actuators. The users perceived the pressure clearly, and the results suggested that the device was sufficiently useful in providing directions. In addition, we were able to direct the user’s attention in an arbitrary direction. The haptic sensation provided by the pneumatic actuators is very simple, and the user can imagine the haptic sensation. In addition, by adding the element of “direction” to the haptic sensation, the user can add the meaning of “space” to the presented haptic sensation. This means that the user can perceive the presented haptic sensation in relation to the environment around them. Therefore, the user can recognize that something exists. In addition, haptic stimuli can induce animacy perception [37,38]. By reproducing the physiological periodic motions unique to living things as haptic sensations, it is possible to provide animacy perception to the user. Therefore, by presenting haptic stimuli with the animacy perception such as the heartbeat in an arbitrary direction, the user can experience the sensation of guidance by living things. We consider that this will lead to an alleviation of the feeling of loneliness. The device can be further improved to offer a greater variety of haptic presentations. For example, increasing the number of actuators will make it possible to present sequential haptic sensations on the user’s neck. We also believe that increasing the contact area between the actuators and the user’s neck will result in a clearer haptic presentation. However, the majority of the subjects felt uncomfortable wearing the device, an issue that must be addressed in the future. In the haptic presentation experiment, no difference in haptic stimulation was observed when differently shaped pneumatic actuators were used. If the actuator shape does not affect the accuracy of direction presentation, users will enjoy the freedom of designing wearable items (e.g., clothing) based on the shape of an actuator. In other words, the proposed system can guarantee its designability as a fashion item.

## 7. Limitations and Future Work

Recognition by the user of the pressure of the pneumatic actuator and its direction means that the user can pay attention to the direction. In other words, the proposed haptic presentation device can determine a person’s position in real-time. The proposed device provides the user with an opportunity to determine the location of a person or an object. Moreover, by presenting a remote user’s specific daily activities as haptic sensations, the wearer of a device can feel a connection with the user. The proposed device is also expected to function as a communication tool, wherein the intensity and pattern of the haptic presentation can be changed depending on the sender of the information or depending on the message. Moreover, the actuator can be used as a communication tool through a haptic presentation by assigning various motions of the actuators to indicate emotions, such as joy or sadness, or to indicate physical states, such as drowsiness or hunger. This approach might enable users to accurately perceive the message indicated by a pressure presentation. However, these functions are not yet implemented, and the current proposed device cannot use these functions.

Future works will require automating the operation of pneumatic actuators. The inflation of the actuators cannot be changed automatically because the actuators are manually operated using the Wizard of Oz method. When actuators are automated, the strength, speed, and frequency of actuator inflation can be finely controlled. Moreover, when the number of actuators is increased, complex and diverse movement patterns may be executed, and the direction indicated by a pressure pattern and what it means can be analyzed more easily. This enables the presentation of new pressure patterns involving sequential inflation of actuators. In addition, to attach an automatic pneumatic actuator to a piece of clothing, it is necessary to design a compact pneumatic circuit (e.g., Bubble [39]). The combination of a micropump, a small solenoid valve, and a microcontroller can realize a device about the size of the palm of the hand, reducing the burden on the part of the user by rendering a device lighter and smaller.

It is necessary to provide external sensors for situations that require interaction with people and objects. For example, navigation to a destination requires GPS and orientation sensors. Recognizing obstacles and maintaining social distance requires cameras and infrared sensors. Thus, utilizing the proposed system in various applications requires a variety of devices for each application. In the future, we intend to investigate the combination of the proposed device and external sensors.

The actuator used in this study consists of two rectangular plastic films layered together, and it inflates from the center. Controlling the manner by which the actuator inflates could make a difference in the haptic sensation felt by the user. For instance, an actuator may be designed in such a way that the center of the actuator becomes more raised or that the actuator is not raised in the center, but only along the periphery. Therefore, we must investigate the differences in user’s haptic perception as a function of actuator inflation in order to provide clearer and more varied haptic presentations. We also believe that improving actuators and devices based on the survey results may improve wearing comfort.

## Figures and Tables

**Figure 1 sensors-23-00584-f001:**
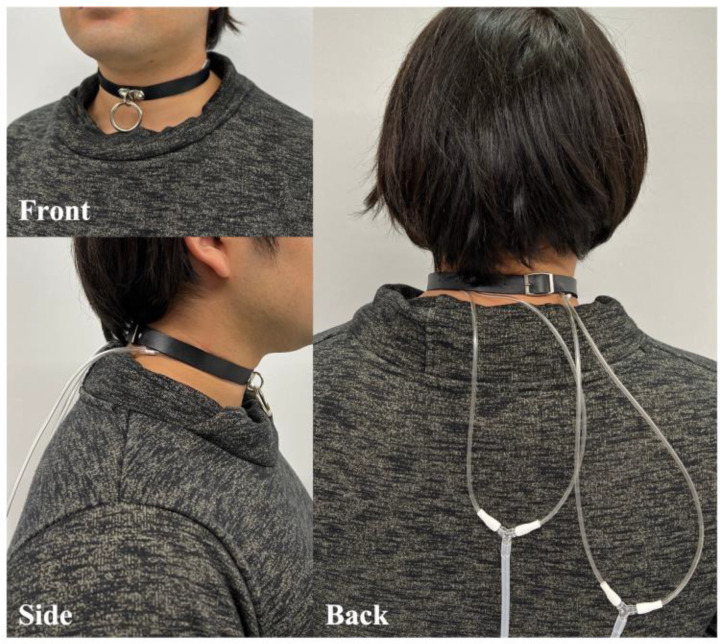
The choker-type haptic presentation device equipped with pneumatic actuators.

**Figure 2 sensors-23-00584-f002:**
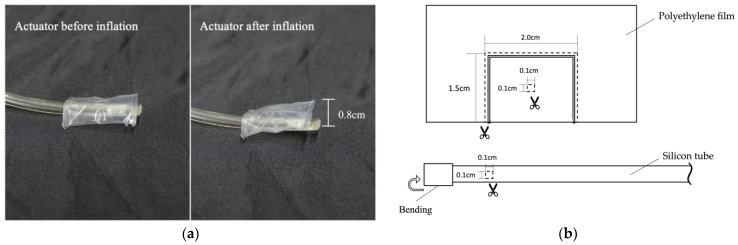
Fabricated pneumatic actuator. (**a**) A working fabricated pneumatic actuator. (**b**) Blueprint of the pneumatic actuator.

**Figure 3 sensors-23-00584-f003:**
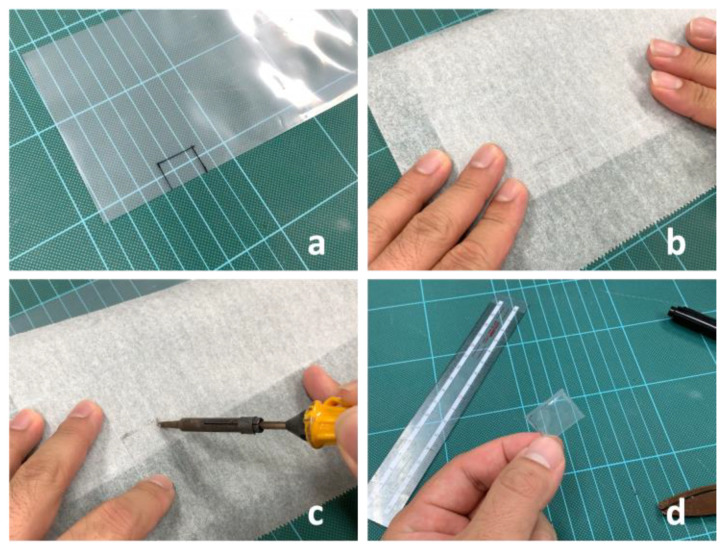
Fabrication procedure for pneumatic actuators. A rectangular outline was drawn on a poly-ethylene film (**a**), then a cookie sheet was placed over the film (**b**). The rectangular outline was traced over the cookie sheet with a soldering iron (**c**), and then the welded area was cut out (**d**).

**Figure 4 sensors-23-00584-f004:**
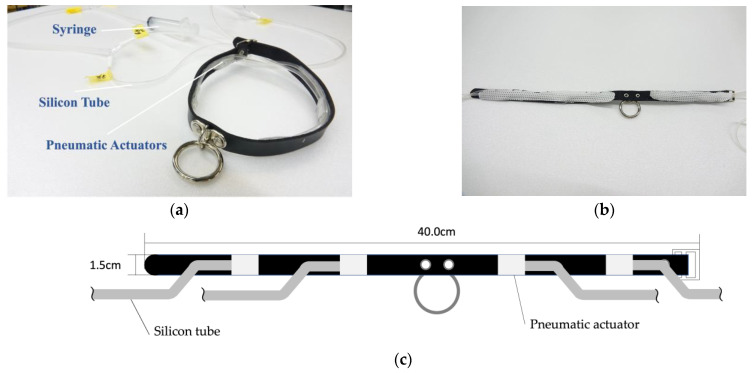
The prototype and the schematic of the haptic presentation device. (**a**) Prototype of the haptic device. (**b**) The actuators and the tubes were concealed by a cloth. (**c**) Blueprint of the proposed device.

**Figure 5 sensors-23-00584-f005:**
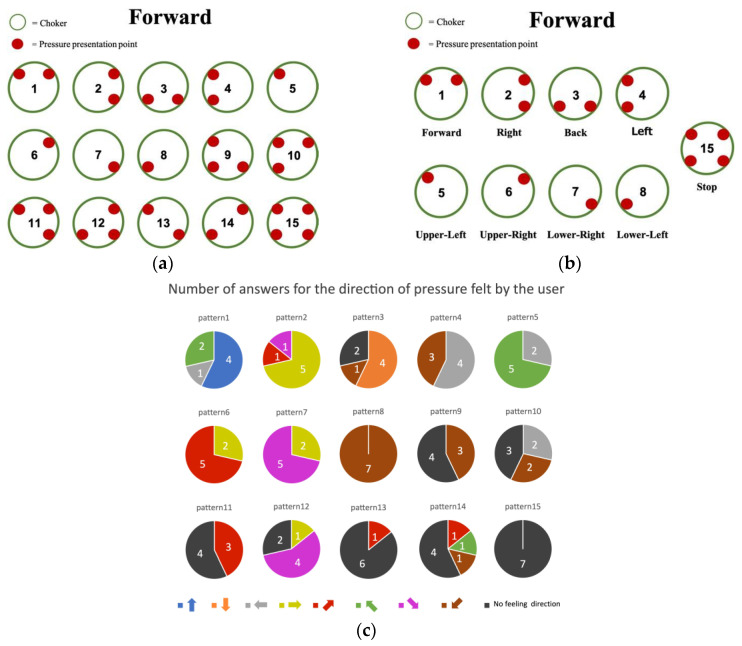
Pressure patterns the proposed device can present. The number in the graph is the number of subjects who responded in that direction. (**a**) Pressure patterns used in the preliminary study. (**b**) Pressure patterns used to indicate directions in the user study. (**c**) Results of Preliminary Study. The numbers in the graph indicate the number of answers. For each direction, the pattern with the highest number of answers was assigned that direction.

**Figure 6 sensors-23-00584-f006:**
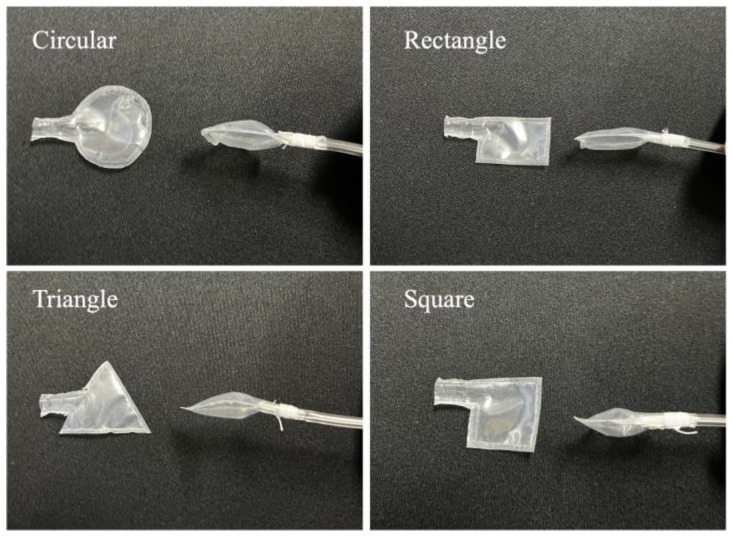
Four shapes were used in the pressure presentation experiments. In each figure, the left side shows the shape of the actuator (side view), and the right side shows the inflated actuator (top view).

**Figure 7 sensors-23-00584-f007:**
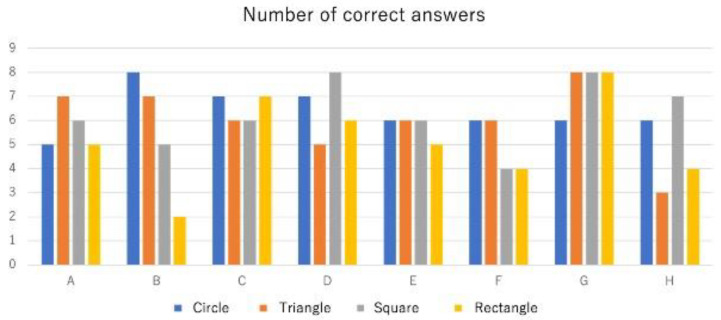
Graphical representation of the distribution of correct answers.

**Figure 8 sensors-23-00584-f008:**
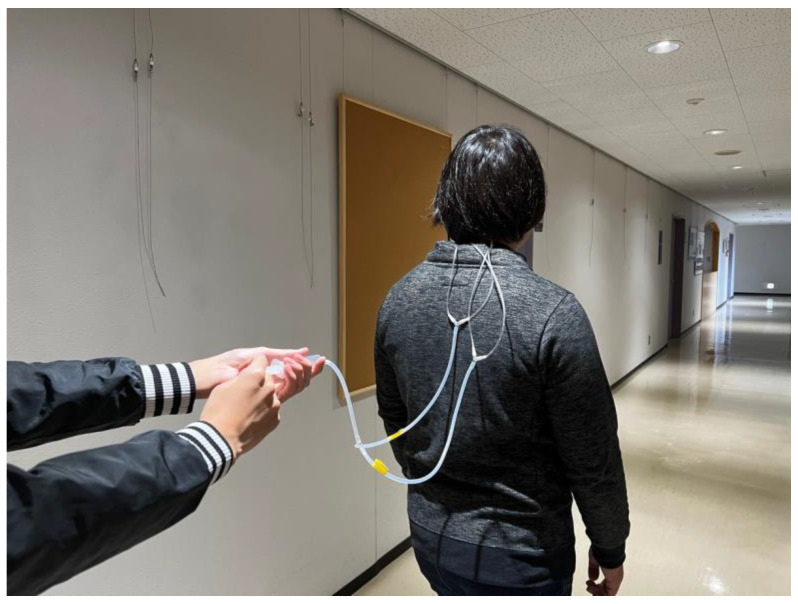
A participant in the user study.

**Figure 9 sensors-23-00584-f009:**
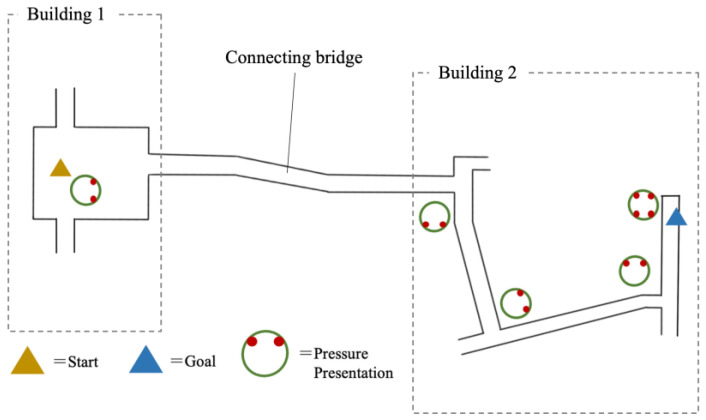
Walking route and the spots where pressure is presented in the user study.

**Figure 10 sensors-23-00584-f010:**
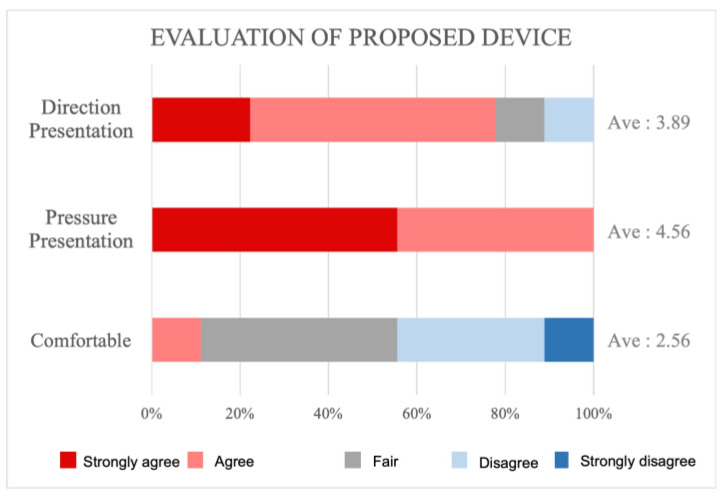
Survey results.

**Table 1 sensors-23-00584-t001:** Comparison of the proposed device with some of the related work introduced. It shows the similarities and differences between them.

	Ours	xClothes [22]	ActiveBelt [25]	Sticky Actuator [27]	Force Jacket [30]	HeadBlaster [31]	Cross-Field Haptics [33]
Safety	✔︎	✔︎	✔︎	✔︎	✔︎	✘	✘
Wearable	✔︎	✔︎	✔︎	✘	✔︎	✔︎	✘
Various applications	✔︎	✘	✔︎	✔︎	✔︎	✘	✘
Low cost	✔︎	✔︎	✘	✔︎	✘	✘	✘
Clear haptic	✔︎	✘	✘	✘	✔︎	✔︎	✔︎

**Table 2 sensors-23-00584-t002:** The table of the number of correct identification of which actuators were inflated. Scores were assigned based on an eight-point scale.

Shape\User	A	B	C	D	E	F	G	H	*M* Total	*F* Total	Total	*M* _ave_	*F* _ave_	*Ave*
Circular	5	8	7	7	6	6	6	6	27	24	51	6.750	6.000	6.375
Triangle	7	7	6	5	6	6	8	3	25	23	48	6.250	5.750	6.000
Square	6	5	6	8	6	4	8	7	25	25	50	6.250	6.250	6.250
Rectangle	5	2	7	6	5	4	8	4	20	21	41	5.000	5.250	5.130
Total	23	22	26	26	23	20	30	20	97	93	190			

**Table 3 sensors-23-00584-t003:** Statistical analysis results showing that no significant difference exists between the two actuator shapes that had a large difference in terms of the number of correct answers.

*C* _ave_	6.375
*R* _ave_	5.125
*z*	1.556
*P*	0.119

**Table 4 sensors-23-00584-t004:** Statistical analysis results showing that no significant gender differences exist in the obtained data.

Shape	Circular	Triangle	Square	Rectangle
*F* _ave_	6.000	5.750	6.250	5.250
*M* _ave_	6.750	6.250	6.250	5.000
*Ave*	6.375	6.000	6.250	5.125
*z*	1.239	0.298	0.149	0.146
*P*	0.215	0.766	0.882	0.884

## Data Availability

The data presented in this study are available on request from the corresponding author.

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
