# Peer review of "Omnidirectional Haptic Stimulation System via Pneumatic Actuators for Presence Presentation"

_sensors, 2023, doi:10.3390/s23020584_

Round 1

Reviewer 1 Report

In this article, the authors have developed a choker-type wearable device that presents various pressure patterns around the neck.  Also, they claimed that, the device can be used in a wide range of applications, from those necessary in daily living to those that enhance one’s experience in the realm of entertainment.

The article is very technically sound and more informative. I have noticed one thing regarding distribution. Therefore, my question is such distribution is what type? Normal, Uniform, Chi square etc. If so, kindly mention in the paper. (In section 4.2).

The paper is very interesting, Innovative and well-written.

The paper may be accepted for publication.

Author Response

Thanks very much for your comments. Please see the attachment for authors responses.

Reviewer 2 Report

The manuscript presents a good structure, description of methodology and description of experiments. However, at many times I had doubts whether the citations corresponded to the bibliographical references made available in the text, since these were not numbered, according to the citations in the text.

Related Works – it would be more helpful to include a table that clearly presents the functional similarities and differences of existing prototypes or designs, cited throughout the manuscript...

What would be the justification for using the prototype on the neck and not on the wrist?

It would be interesting to evaluate the level of comfort and usability of the prototype.

Author Response

Thanks very much for your comments. Please see the attachment for author's response.

Reviewer 3 Report

The manuscript is dedicated to study of a choker-type haptic presentation device driven by a set of pneumatic actuators where different shapes of actuators, haptic sensations, directional functionalities as well as safety and comfort are discussed. The manuscript is basically well-written and sound; however, a few ambiguous issues may need further clarification. 

1. Regarding the main statement of study (1st sentence, paragraph 4, page 2), the authors aimed to investigate the effectiveness of the proposed device in relieving loneliness. It is seemly not so clear about how they could achieve this goal in the manuscript. Please explain it.

2. In the same paragraph above, the authors claimed that the proposed device has the capability of reproducing someone’s heartbeat info to another. However, this point is not presented in the manuscript. Authors should address this claim with sufficient evidence, if any.  

3. In second part of Summary, the authors blended their outcomes with the expectations/perspectives together, which may mislead the readers.     

Author Response

Thanks very much for the comments. Please see the attachment for authors' comments.

Round 2

Reviewer 2 Report

The manuscript presents a clear approach to the prototype and experiments.

I believe that all observations have been answered.

Reviewer 3 Report

I am broadly happy with the revised manuscript and also very pleased to suggest the Editor to accept this manuscript in present form.